# Destruxin A Interacts with Aminoacyl tRNA Synthases in *Bombyx mori*

**DOI:** 10.3390/jof7080593

**Published:** 2021-07-23

**Authors:** Jingjing Wang, Alexander Berestetskiy, Qiongbo Hu

**Affiliations:** 1Key Laboratory of Bio-Pesticide Innovation and Application of Guangdong Province, College of Plant Protection, South China Agricultural University, Guangzhou 510642, China; wangjingjing@scau.edu.cn; 2Department of Phytotoxicology and Biotechnology, All-Russian Institute of Plant Protection, Shosse Podbelskogo, 3 Pushkin, 196608 Saint-Petersburg, Russia; aberestetskiy@vizr.spb.ru

**Keywords:** destruxin, aminoacyl tRNA synthetase, silkworm, binding protein, binding model

## Abstract

Destruxin A (DA), a hexa-cyclodepsipeptidic mycotoxin produced by the entomopathogenic fungus *Metarhizium anisopliae*, exhibits insecticidal activities in a wide range of pests and is known as an innate immunity inhibitor. However, its mechanism of action requires further investigation. In this research, the interactions of DA with the six aminoacyl tRNA synthetases (ARSs) of *Bombyx mori*, BmAlaRS, BmCysRS, BmMetRS, BmValRS, BmIleRS, and BmGluProRS, were analyzed. The six ARSs were expressed and purified. The BLI (biolayer interferometry) results indicated that DA binds these ARSs with the affinity indices (K_D_) of 10^−4^ to 10^−5^ M. The molecular docking suggested a similar interaction mode of DA with ARSs, whereby DA settled into a pocket through hydrogen bonds with Asn, Arg, His, Lys, and Tyr of ARSs. Furthermore, DA treatments decreased the contents of soluble protein and free amino acids in Bm12 cells, which suggested that DA impedes protein synthesis. Lastly, the ARSs in Bm12 cells were all downregulated by DA stress. This study sheds light on exploring and answering the molecular target of DA against target insects.

## 1. Introduction

Destruxins are secondary metabolites secreted by entomopathogenic fungus *Metarhizium anisopliae*, and they have insecticidal effects by destroying innate immunity and accelerating pathogenesis [1,2]. They are synthesized by a cluster of genes including non-ribosomal peptide synthetase and, therefore, have a series of analogues. Among them, destruxin A (DA) is more common and shows better insecticidal activity in infection. Structurally, destruxins are hexa-cyclodepsipeptides condensed and cyclized by proline, isoleucine, *N*-methyl-valine, *N*-methyl-alanine, β-alanine, and α-hydroxyl acid. As a considerable insecticidal mycotoxin, DA was reported to affect muscle contraction [3], suppress hemolymph immunity effects, and induce ion equilibrium chaos in the hemocytes of insects [4,5]. This compound has potential as a new insecticide [6,7]. Although several DA-binding proteins have been found [8,9] in *Bombyx mori*, the molecular mechanism underlying DA entry into insect tissues or cells is still obscure.

Previously, our proteomics experiments, termed DARTS (drug affinity responsive target stability), revealed that six cytoplasmic aminoacyl tRNA synthetases (ARSs) were considered as DA-binding protein candidates in *B. mori* Bm12 cells. ARSs are indispensable enzymes that catalyze the connection of amino acids to the 3′ ends of their cognate tRNAs in protein translation [10]. There are three steps in the aminoacyl-tRNA synthesis reaction. An amino acid is activated by an ATP molecule to form the amino acid–AMP intermediate in the active site of the corresponding ARS; then, the cognate tRNA binds to the ARS through its anticodon-binding domain; lastly, the amino acid is transferred to the tRNA [11]. In addition to their canonical role in protein synthesis, ARSs were shown to be involved in a wide range of physiological regulation include supporting RNA splicing, cell signaling, and transcriptional and translational regulation [12]. Actually, ARSs have been researched thoroughly as therapeutic targets of bacterial and parasitic infection and other human diseases [13]. However, there have been few studies on ARSs in silkworms.

Owing to its special structure, cyclopeptide DA was proposed to interact with parts of ARSs in *B. mori* according to our preceding analysis; moreover, we intriguingly verified that arginyl tRNA synthetase (ArgRS) is a DA-binding-protein [14]. Here, we investigate the interactions between DA with six silkworm ARSs (BmAlaRS, BmCysRS, BmMetRS, BmValRS, BmIleRS, and BmGluProRS). This research might provide more detailed clues about the DA molecular mechanism underlying the destruction of the intracellular steady state in silkworm.

## 2. Materials and Methods

### 2.1. Cell Lines and Compounds

The *Bombyx mori* ovary-derived Bm12 cell line was donated by Professor Cao Yang (South China Agricultural University) and cultured in Grace’s insect medium (Gibco^TM^, Waltham, MA, USA) and 10% fetal bovine serum (Gibco^TM^, Waltham, MA, USA). Cells were cultured in a constant-temperature incubator at 27 °C with specific humidity. Logarithmic-phase cells were used for the experiment. All experimental cells were passed within 50 generations. 

Destruxin A (DA) was isolated and purified from *Metarhizium anisopliae* var. *anisopliae* strain MaQ10 [15] in our laboratory and stored at −80 °C. It was dissolved using dimethyl sulfoxide (DMSO) as a stock of 10,000 mg/L and diluted with medium to the working concentration for the experiment.

### 2.2. Preparation of Protein and Biolayer Interferometry (BLI)

All experimental proteins were His-tagged, expressed in bacterium *E. coli*, and purified by nickel affinity chromatography. The protein accession numbers of BmAlaRS, BmCysRS, BmMetRS, BmValRS, BmIleRS, and BmGluProRS are NP_001037452.1, XP_004927917.2, XP_012544872.1, XP_004923659.1, XP_004929693.1, and XP_012544610.1 respectively. The purity and concentration of all experimental proteins were satisfied for BLI.

BLI assessment was performed on a ForteBio OctetQK System (K2, Pall Fortebio Corp, Menlo Park, CA, USA). Firstly, the protein sample was immobilized to a special biosensor. Secondly, serial gradient dilutions of molecules were prepared using PBST buffer (0.05% Tween20, 5% DMSO). This buffer was also used throughout the test. The instrument detected the light signal during molecule flowthrough. The working procedure involved a baseline for 60 s, association for 60 s, and dissociation for 60 s. The raw data were collected and processed using Data Analysis Software (9.0, Pall Fortebio Corp, Menlo Park, CA, USA). 

### 2.3. Homology Modeling and Molecular Docking

The template crystal structures for ARSs were identified through BLAST and downloaded from the RCSB Protein Data Bank. The templates of BmAlaRS, BmCysRS, BmMetRS, BmValRS, BmIleRS, and BmGluProRS are 4XEM, 1LI5, 5GL7, 1GAX, 1ILE, and 4HVC, respectively. Homology modeling was conducted using MOE (Chemical Computing Group, Montreal, QC, Canada). 

MOE Dock was used for molecular docking of ARSs with the small molecule DA, as well as predicting the binding affinity. Detailed steps can be found in our previous study. 

### 2.4. Assessment of Gene Expression

Expression of the target gene was measured by RT-qPCR. The qPCR reactive program was subjected to 39 cycles at 95 °C for 10 s, 60 °C for 10 s, 72 °C for 30 s, 95 °C for 10 s, and 65–95 °C for 5 s. The experiment was repeated three times. The silkworm GAPDH (glyceraldehyde-3-phosphate dehydrogenase) gene was taken as the reference gene. The qPCR data were analyzed by using the 2^−ΔΔCt^ method (Table 1).

### 2.5. Detection of Soluble Protein and Free Amino-Acid Content

After being treated with DA, the Bm12 cell lines were collected and washed with PBS, followed by lysing with a freeze–thaw cycle. The supernatants were collected as soluble protein and used for STabeDS-PAGE. DA-treated Bm12 cell line supernatants were collected and precipitated by sulfosalicylic acid. The supernatants were collected as a free amino-acid mix and analyzed using a Hitachi L-8900 (Tokyo, Japan).

## 3. Results

### 3.1. Interaction of DA with Six Silkworm ARSs

Recombinant ARSs proteins were generated by *Escherichia coli* and purified by nickel affinity chromatography. Affinity assessment was conducted using the optic-based method biolayer interferometry (BLI). Interestingly, the BLI results indicated that all test silkworm ARSs interacted with DA, and the affinity values ranged from 10^−4^ to 10^−5^ M level (Table 2, Figure 1). In detail, the affinity values (K_D_) of BmAlaRS, BmCysRS, BmMetRS, BmValRS, BmIleRS, and BmGluProRS were 3.18 × 10^−4^, 7.12 × 10^−5^, 6.61 × 10^−5^, 4.39 × 10^−5^, 5.90 × 10^−4^, and 6.04 × 10^−4^ M, respectively.

### 3.2. Analysis of Soluble Protein, Free Amino Acids, and Gene Expression Levels of ARSs in Bm12 Cells

Our previous study showed that DA might impede protein synthesis; therefore, we analyzed the changes in soluble protein content in Bm12 cell lines and identified a negative time-dependency of DA treatments with concentrations ranging from 10 μg/mL to 50 μg/mL (Figure 2A). The results indicated that DA inhibited protein synthesis at a 10–20% level in 24 h. 

Owing to several ARSs exhibiting high affinity with DA, we assumed that the inhibition of ARS activity would accumulate free amino acids in cells (Figure 2B). The results indicated that, after treatment with DA for 3–6 h, the relative amino-acid content increased, before remaining relatively steady. Furthermore, there were no significant DA dosage effects on the free amino-acid content. The content of cysteine seemingly changed to a greater extent than other amino acids, which might be related to BmCysRS being the strongest binding protein of DA. 

We then investigated the expression level of the six ARSs under dosage- and time-dependent DA treatments, and the results showed that all ARSs were downregulated after DA stress (Figure 3). There is no obvious correlation between the expression of each ARS and continuous DA concentration and time. BmCysRS, BmMetRS, and BmGluProRS were relatively highly expressed, whereas BmValRS had relatively low expression.

### 3.3. Interaction Mode of DA with Silkworm ARSs

To investigate the binding affinity of DA with ARSs, docking simulation studies were carried out. The docking scores are shown in Table 3 and ranged from −7.49 to −9.23 kcal/mol. 

The binding modes between DA and ARSs are shown in Figure 4. DA formed a suitable steric complementarity with the binding sites of all ARSs. Furthermore, hydrogen bond interactions were formed between DA and ARSs. In detail, as a hydrogen bond acceptor, the oxygen atoms of DA formed hydrogen bonds with the side-chains of Arg77 and Arg246 in BmAlaRS, with the side-chains of Arg1177 and Arg1188 in BmGluProRS, with the side-chain of Lys93 in BmCysRS, with the side-chain of His58 in BmIleRS, with the backbone of Tyr266 and side-chain of Asn269 in BmMetRS, and with the backbone of Asn131 and side-chain of Lys642 in BmValRS. 

The sizes of the binding pockets of the above ARSs, as well as the surrounding residues, were similar to each other. The steric complementarity of DA with the pockets of these ARSs was similar. The electrostatic matching was only slightly different; thus, the binding free energy (affinity) was similar.

## 4. Discussion

In order to determine the molecular target of DA in insect tissues or cells, we conducted a series of proteomics experiments in silkworm to identify its binding proteins. In all, six ARSs were isolated as candidates in Bm12 cells but not in hemolymph, and further experiments verified them as DA-binding proteins. ARSs are significant enzymes in living beings, with a canonical role in catalyzing amino acid and tRNA linkage during protein translation; more importantly, they are involved and regulated in a wide range of physiological processes [10,16]. ARSs are therapeutic targets in antimicrobials on account of their structures being different in bacteria, parasites, and humans, with most inhibitors binding to their active site [17]. However, studies of silkworm ARSs are rare in functional research and structural identification, which absolutely impedes a determination of the further effects that DA might induce. Thus, we employed modeling and docking to theoretically and structurally elucidate these connections. On account of distribution, ARSs are divided into cytoplasmic and mitochondrial sets [18], and the six silkworm ARSs we isolated from proteomics were all cytoplasmic type. 

Interestingly, DA interacts with this family of enzymes with similar affinity. It is worth noting that DB (destruxin B) also interacts with the above ARSs showing the same affinity as DA (data not shown). They are structurally similar analogues with differences in the vinyl and ethyl groups, whereby DB is further oxidated to DA in the synthesis process. Intriguingly, depending on our preceding experiments, their binding proteins were different, and the mechanism of DB differed from that of DA, especially in ATPase, where DB exhibited 100-fold improved affinity compared to DA in hemocytes [5]. In fact, owing to this slight difference, DB shows less insecticidal activity and larger antitumor activity than DA [19,20]. As for ARSs, the hypothesis for the same affinity of DA and DB is that the backbone of destruxins, containing amino-acid residues, mimics the substrates of ARSs. The modeling results revealed that the difference in DA and DB did not affect the complex interaction.

From the evolution aspect, it is reasonable that destruxins show no selectivity and not so high affinity to ARSs in host insects. *M. anisopliae* secretes insecticidal destruxins when infecting the host; its aim is not to kill the host but to gradually break the host immune defense to facilitate better colonization and increased spore production. Destruxins might bind to host insects’ ARSs with no selectivity and inhibit their arginyl-tRNA synthesis so as to suppress protein synthesis, especially immune stress proteins. Interestingly, several DA-binding-proteins we previously found were at the millimolar level, and we speculate that DA adopts a multitarget strategy at the millimolar level to injure the host to a greater extent.

Previously, we realized that DA might inhibit the protein synthesis pathway. Here, we confirmed this by the time-dependent decrease in soluble protein and free amino-acid accumulation in early treatments. Meanwhile, the interaction of DA with other elements in protein synthesis is still being studied by our group. When under stress, cells might stall translation and protein synthesis for the sake of energy saving [21], which could explain why ARSs are downregulated upon DA exposure. Although ARSs are DA-binding proteins, it seems that DA impacts the upstream gene of ARSs before their binding. Studies have shown that, beyond their catalytic role in protein synthesis, ARSs also contribute to the immune response under stress [22]. Interestingly, the bifunctional BmGluProRS was reported to exhibit an immune response in other species, and it had a high frequency of occurrence in our previous proteome experiments, which indicated that it is not only a DA-binding protein but it also contributes to the immune response under DA stress, as reflected by the relatively high expression (Figure 3).

## 5. Conclusions

Here, we investigated the interactions of six cytoplasmic silkworm aminoacyl tRNA synthetases with insecticide mycotoxin DA in terms of their affinity and interaction mode. Gene expression experiments showed that ARSs are not direct targets of DA. In conclusion, DA interacts with BmAlaRS, BmCysRS, BmMetRS, BmValRS, BmIleRS, and BmGluProRS proteins at 10^−4^ to 10^−5^ M affinity, exhibiting the same interaction mode. At the millimolar level, we speculate that DA adopts a multitarget strategy to disrupt the protein synthesis pathway.

## Figures and Tables

**Figure 1 jof-07-00593-f001:**
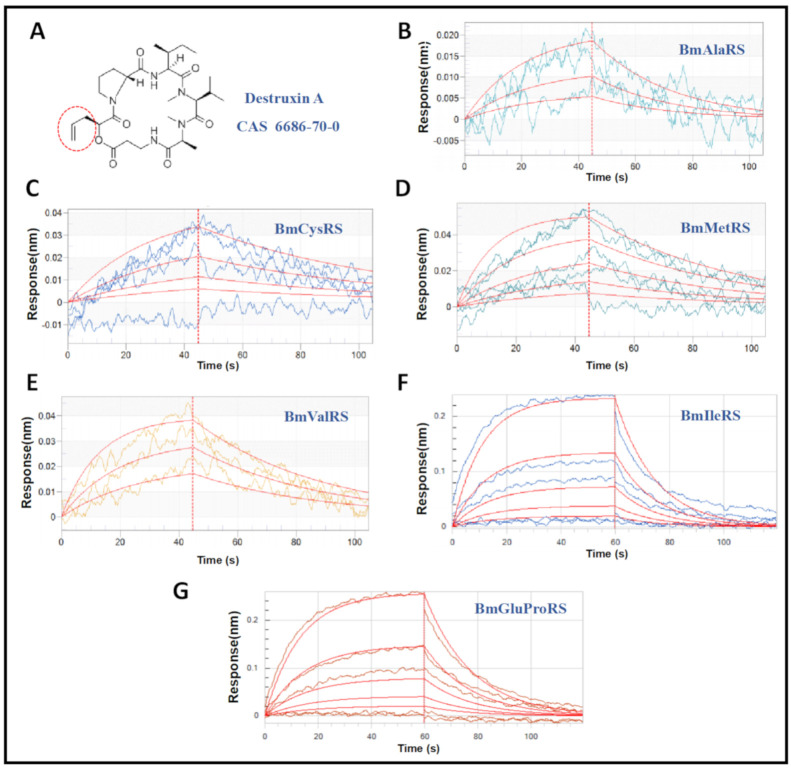
Profiles of interaction between destruxin A (DA) and ARSs. (**A**) Structure of DA. (**B**–**G**) Detailed interaction analyses of DA with BmAlaRS, BmCysRS, BmMetRS, BmValRS, BmIleRS, and BmGluProRS, respectively.

**Figure 2 jof-07-00593-f002:**
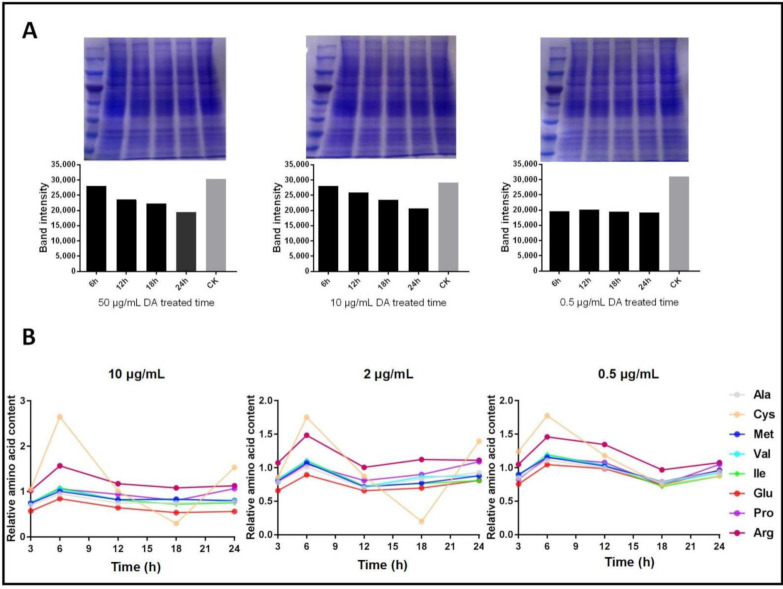
Effects induced by the binding of DA with ARSs in Bm12 cell lines. (**A**) DA influences soluble protein content in a negatively time-dependent manner. (**B**) DA treatment initially causes the accumulation of free amino acids.

**Figure 3 jof-07-00593-f003:**
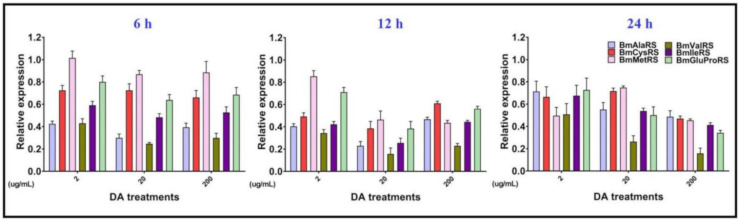
Analysis of effects of silkworm ARS gene expression in Bm12 cells after DA dosage and time treatments.

**Figure 4 jof-07-00593-f004:**
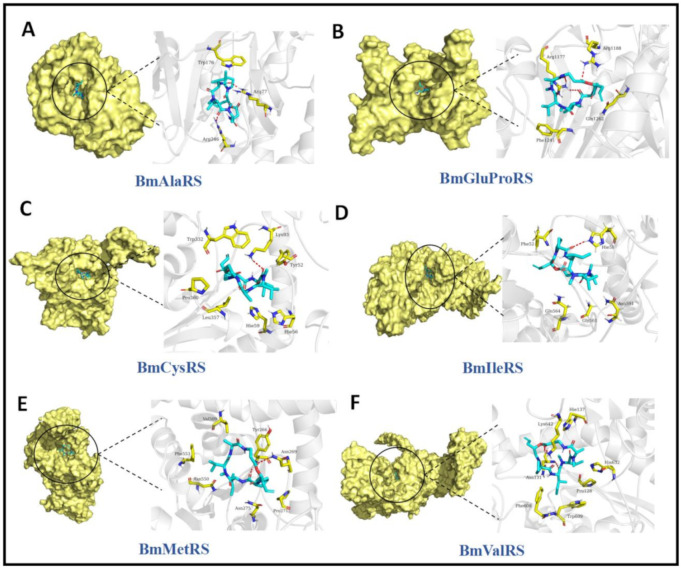
Complexes of ARSs and DA. The left part of each image is the binding mode of DA with ARSs shown in molecular surface representation. DA is colored cyan, while the molecular surface of ARSs is colored pale yellow. The right part of each picture is the 3D binding mode of ARSs and DA. DA is colored cyan, the surrounding residues in the binding pockets are colored yellow, and the backbone of the receptor is depicted as a transparent white cartoon. (**A**–**F**) The interaction pocket of BmAlaRS, BmGluProRS, BmCysRS, BmIleRS, BmMetRS and BmValRS with DA respectively.

**Table 1 jof-07-00593-t001:** Primers of the silkworm ARSs genes for RT-qPCR detecting.

Gene	Primer (5′→3′)
BmAlaRS	F: ACATGGCGTATCGTGTCTTGGC
	R: TCTGACGCATATCGCACAGCTC
BmCysRS	F: ACGCCACACCGCAAGACAAC
	R: CCATCCACCGCCGCAACC
BmMetRS	F: TCTGGCTGACAGGTTCGTGGAG
	R: CCGCACTTGTCGCACTGGTC
BmValRS	F: ACCTGTGGCTCTACGACCTGTG
	R: CGGTTACGAACGGCATGAAGGG
BmIleRS	F: AGGGGCGCGTTTGAAAGGTG
	R: CTGGTCACTGCCTGTAGCTTGG
BmGluProRS	F: AGCAACCGCAACCTTTCCTGAG
	R: CGCCGCCAGCAAACTTCTCC
GAPDH	F: ATGTTTGTTGTGGGTGTTA
	R: GTAGAGGCAGGAATGATT

**Table 2 jof-07-00593-t002:** Detailed data of BLI assessment.

Proteins	K_on_ (1/M·s) ^1^	K_dis_ (1/s) ^2^	K_D_ (M) ^3^
BmAlaRS	1.15 × 10^2^	3.66 × 10^−2^	3.18 × 10^−4^
BmCysRS	2.10 × 10^2^	1.49 × 10^−2^	7.12 × 10^−5^
BmMetRS	3.07 × 10^2^	2.03 × 10^−2^	6.61 × 10^−5^
BmValRS	5.72 × 10^2^	2.32 × 10^−2^	4.39 × 10^−5^
BmIleRS	1.15 × 10^2^	6.80 × 10^−2^	5.90 × 10^−4^
BmGluProRS	9.39 × 10	5.67 × 10^−2^	6.04 × 10^−4^

^1^ K_on_: association rate constant; ^2^ K_dis_: dissociation rate constant; ^3^ K_D_: affinity constant.

**Table 3 jof-07-00593-t003:** The docking scores of DA with ARSs.

Proteins	Docking Free Energy (kcal/mol)
BmAlaRS	−8.87
BmGluProRS	−7.78
BmCysRS	−8.97
BmIleRS	−7.49
BmMetRS	−8.25
BmValRS	−9.23

## Data Availability

All relevant data are included within the manuscript.

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
