# Peer review of "Destruxin A Interacts with Aminoacyl tRNA Synthases in Bombyx mori"

_jof, 2021, doi:10.3390/jof7080593_

Round 1
Reviewer 1 Report
Reviewer comments
Title: Destruxin A interacts with Aminoacyl tRNA Synthases in Bombyx mori
Destruxin A interacts with Aminoacyl tRNA synthasesn on silkworm is an interesting result and straight forward study. Up to my knowledge it is a new study and added information on the available literature
However, there are some major pointes to be rectified in order to make the manuscript for further consideration. Reference section should be updated and discuss more about microbial pesticide in the introduction as well in the discussion at least the author should refer the recent documents and rewrite the advantages of microbial pesticide. Please refer several articles published recently in Journal of Fungi.
Abstract
L10-11 Genus and species name should be italicised
L12- the sentence ‘But its mechanism of action is not clear yet’ shall revised as ‘ But its mechanism of action is still under investigation’
L22 -Key word dose not duplicate what the authors presented in the heading
L25 What is meant by ‘Destruxin’. The authors should know why the work was done. Introduction section should be included about the advantages of microbial pathogen and Destruxin. please do refer the previously published papers doi: 10.1016/j.jip.2018.10.008; doi:10.1007/978-81-322-2056-5_3; doi: 10.1080/03235408.2013.840999. and also specify the relation between Destruxin and all the endomopathogenic fungi with relevant supporting research.
L64 dimethyl sulfoxide (DMSO) How much concentration did you use?
L77 sample was immobilized to special biosensor.What type of biosensor?
L105-What about statistical programme?
L146 Figure 3. Analysis of effects of silkworm ARSs gene expression in Bm12 cells after DA dosage and time treatments. I can’t relay without statistics…need to analysis the data
L174 Discussion need to concertante your work in relation with other work done by the researchers. Please do discuss properly with recent citation. I am not convinced with current discussion as well as reference Whole result, discussion and conclusion section needs to develop according to your result and appropriate citation must be provided to justify your claim
Reviewer 2 Report
The works in this manuscript are interesting. Several questions for authors:
(1) English needs to be improved. line 10: Metarhizium anisopliae need to be italic. line 37 the full name for DARTS. line 65 italic name for fungi name. line 70: E. coli need to be italic. line 107: Sis silkworm. what is Sis? line 108: E. coli. The description in the text needs to be refined, not very smooth.
(2) introduction part: in second paragraph, please put citation in line 39 when you mention your previous study.
(3) materials and methods: (a) line 67: what is the concentration of DMSO you used for dissolve DA? or I should say: when you treated your Bm12 cell with DA, what the percentage of DMSO in your cell culture? (b) section 2.4. please state reverse transcription reaction. Additionally, what the PCR efficiency for your PCR primers? please add this information in your Table 3. (c) section 2.5. line 103: what do you mean the other part supernatant? and for what propose when you do precipitation by sulfosalicyclic acid? I can not make sense with this. Please state more clearly about your amino acid analysis.
(4) results section. (a) In figure 1 legend, why you mention DB? not necessary to mention this, can be deleted. Correct GlnPro to GluPro. (b) section 3.2. you do not need to describe the result of 0.5 ug/mL DA treatment (line 126). Regarding your fig 2, it is not convinced me with these data since you did not state how much cell vol (or cell number) used for this gel analysis. Can you do BCA protein quantification? (c) regarding fig.3, this format is hard for readers to compare data between treatments. Since you have time course, can you use line chart to draw or other graph to present? Additionally, the description in text is so less (just 4 lines, line 139-143). Furthermore, it is no sense to detect ARS gene expression, since DA work on ARS protein, not gene. You can detect protein level of immune or other physiology, rather than gene expression.
Round 2
Reviewer 1 Report
The authors revised the manuscript satisfactory and the manuscript can be accepted now.
Reviewer 2 Report
No further comments.